# Impact of a virtual learning environment on the conscious prescription of antibiotics among Colombian dentists

**María del Pilar Angarita-Díaz**[1]*, **Lilia Bernal-Cepeda**[2], **Leidy Bastidas-Legarda**[3], **Diana Forero-Escobar**[1], **Angélica Ricaurte-Avendaño**[4], **Julián Mora-Reina**[1], **Martha Vergara-Mercado**[5], **Alejandra Herrera-Herrera**[6], **Martha Rodriguez-Paz**[7], **Sandra Cáceres- Matta**[8], **Natalia Fortich-Mesa**[9], **Emilia María Ochoa-Acosta**[10]

1 Facultad de Odontología, Universidad Cooperativa de Colombia, Villavicencio, Colombia, 2 Facultad de Odontología, Universidad Nacional de Colombia, Bogotá, Colombia, 3 School of Clinical and Experimental Sciences, Faculty of Medicine, University of Southampton, Southampton, United Kingdom, 4 Sub-Direction of E-learning, Universidad Cooperativa de Colombia, Medellín, Colombia, 5 Facultad de Odontología, Universidad del Sinú, Montería, Colombia, 6 Facultad de Odontología, Universidad Metropolitana, Barranquilla, Colombia, 7 Facultad de Odontología, Universidad del Valle, Cali, Colombia, 8 Facultad de Odontología, Universidad del Sinú, Cartagena, Colombia, 9 Facultad de Odontología, Universidad Rafael Núñez, Cartagena, Colombia, 10 Facultad de Odontología, Universidad Cooperativa de Colombia, Medellín, Colombia

* maria.angaritad@campusucc.edu.co

**Data Availability Statement:** All relevant data are within the paper.

## Abstract

Appropriate antibiotic prescription contributes to reducing bacterial resistance; therefore, it is critical to provide training regarding this challenge. The objective of this study was to develop a virtual learning environment for antibiotic prescription and to determine its impact on dentists' awareness, attitudes, and intention to practice. First, the learning content on multimedia resources was developed and distributed into three challenges that participants had to overcome. Then, a quasi-experimental study was performed in which the virtual learning environment was implemented on dentists from seven Colombian cities. The median of correct answers and the levels of awareness, attitudes, and intention to practice were compared before, immediately after, and 6-months post-intervention. Wilcoxon signed-rank and McNemar's tests were used to determine the differences. A total of 206 participants who finished the virtual learning environment activities exhibited a favorable and statistically significant impact on the median of correct answers of awareness (p < 0.001), attitudes (p < 0.001), and intention to practice (p = 0.042). A significant increase occurred in the number of participants with a high level of awareness (p < 0.001) and a non-significant increase in participants with high levels of attitudes (p = 0.230) and intention to practice (p = 0.286). At 6 months, the positive effect on the median of correct answers on awareness and intention to practice persisted (p < 0.001); however, this was not evident for attitudes (p = 0.105). Moreover, there was a significant decrease in the number of partici-pants who showed low levels of awareness (p = 0.019) and a slight increase in those with high levels of the same component (p = 0.161). The use of a virtual learning environment designed for dentists contributed to a rapid improvement in awareness and intention to

**Funding:** This research has been funded by the Universidad Cooperativa de Colombia (INV2170) (MPA-D, DF-E, JM-R, AR-A, EMO-A), Colgate-ACFO Award 2018, Universidad Nacional de Colombia (45685) (LB-C), Universidad del Valle (MR-P), Universidad del Sinú (009493103032020207) (MV-M, SC-M), Universidad Metropolitana (257-2019) (AH-H) and Corporación Universitaria Rafael Núñez (NF-M). The funders had no role in study design, data collection and analysis, decision to publish, or preparation of the manuscript.

**Competing interests:** The authors have declared that no competing interests exist.

practice antibiotic prescription; however, their attitudes and information retention need reinforcement.

## Introduction

In 2018, 700,000 deaths associated with antibiotic resistance were reported globally including common diseases such as respiratory tract, sexually transmitted, and urinary infections [1]. This number is expected to increase to 10 million deaths annually by 2050, as anticipated by the British government [2]. In 2015, the World Health Organization (WHO) published a global action plan on antimicrobial resistance aimed at inviting all healthcare sectors to contribute to the fight against this multifaceted problem that affects the entire population [3]. Subsequently, the FDI World Dental Federation, through its policy statement on antibiotic stewardship in dentistry, highlighted the critical role of dentists in combating this crisis and encouraged them to prescribe these medicaments rationally [4].

Public health measures to reduce antimicrobial resistance include optimizing antibiotic usage in all healthcare fields [5]. In dentistry, most diseases caused by oral bacteria (dental caries, gingivitis, pulpitis, and periapical, peri-implant, and chronic periodontal infections) only require local intervention; therefore, antibiotic therapy is not needed [6]. However, in some clinical conditions, an antibiotic prescription is essential as a therapeutic or prophylactic measure. In a therapeutic context, antibiotics are prescribed to patients with certain medical conditions that may affect local treatment or as adjuvant therapy after a local intervention in aggressive odontogenic infections that can be life-threatening (such as orofacial abscesses, necrotic periodontal diseases, and pericoronitis) [7]. The American Heart Association (AHA) recommends the use of antibiotics primarily in patients at increased risk of developing infective endocarditis (IE), such as those with certain heart diseases [8]. Prophylactic therapy is not recommended for patients with joint replacement, although the American Dental Association (ADA) indicates that in some cases, the orthopedic surgeon may define the antibiotic regimen [9]. Additionally, the American Association of Endodontics (AAE) suggests that in other health conditions (immunodeficiencies, diabetes, and joint infections), the dentist, physician, and patient should consider the potential risks if a dental procedure is carried out without antibiotic prophylaxis as well as those that can be derived from antibiotic therapy [10].

Studies that have assessed antimicrobial prescription patterns in dentists, through the analysis of awareness, attitudes, and intention to practice or practices, reported unnecessary prescriptions associated with the lack of clarity in antibiotic prescribing guidelines, lack of professional updates, and fear of complications after treatment [11, 12]. *Awareness* is defined as the conscious and personally relevant knowledge on a particular topic [13]; *attitudes* refer to the evaluation of objects of thought that can be observed as stable entities stored in the memory or temporary judgments constructed from information [14], and *intention to practice* is considered as the mental preparation of an individual before defining an action to perform [15, 16]. Together, these three components permit researchers to collect information to design and implement strategies that improve antibiotic prescription practices.

Within the global action plan on antimicrobial resistance developed by the WHO, the first strategic objective is to "improve awareness and understanding of antimicrobial resistance through effective communication, education, and training" [3]. In this regard, some studies have implemented educational interventions aimed at improving antibiotic prescription in dentistry [17], with only a few of them performing in a virtual scenario [18, 19], although it has

been shown that virtual learning environments (VLEs) favor the transfer of knowledge and abilities to a limited number of participants [20]. Some environments use technology with didactic and administrative support needed for the learning process [21]. Moreover, these environments follow an academic curriculum and contain didactic resources as well as systems to track student activities, online support, and other components that seek to simulate a real academic scenario [22].

Didactic virtual resources include multimedia education that uses different tools such as videos, texts, sounds, graphs, and animation [23], presenting dynamic and interactive information that stimulates cognitive processes and facilitates learning. The above is associated with Mayer's cognitive theory of multimedia learning, which states that course design should be aligned to the human cognitive architecture or how human beings process information [24] and comply with certain characteristics such as motivation, stimulation, and information delivery in a simple way with a linear sequence [25]. Considering that knowledge improvement of antimicrobial resistance through robust educational and awareness activities is a priority, the objective of this study was to develop a VLE for appropriate antibiotic prescription in dentistry and to determine its impact on dentists' awareness, attitudes, and intention to practice. Our null hypothesis was that the development and implementation of a VLE on conscious antibiotic prescription in dentistry does not have a favorable impact on participants' awareness, attitudes, and intention to practice.

## Materials and methods

### Study design and study population

This study was approved by the ethics subcommittee of Universidad Cooperativa de Colombia on the 18th of May 2018 (No. 015–2018). Participants provided written consent through the VLE. A quasi-experimental multicenter study with a before-and-after design and without a control group was conducted from March to May 2020 in seven Colombian cities: Barranquilla, Bogota, Cali, Cartagena, Medellin, Monteria, and Villavicencio. Dentists who had been part of our preliminary study on awareness, attitudes, and intention to practice antibiotic prescriptions [26] were invited to participate by researchers in each city. The sample size was calculated for each city, including dentists with a dentistry degree until 2016, according to the last update from the Observatorio Laboral para la Educación through a paired sample t-test (repeated measures). By accepting an alpha risk of 0.05 (5%) and a beta risk of 0.1 (10% potential) in a bilateral contrast, a total of 140 dentists were required. A 15% loss to follow-up was also estimated. At least 20 dentists per city registered at the local Health Secretary or the Sistema Integrado de Información de la Protección Social of the Ministry of Health were required.

### Educational strategy

The VLE was designed in accordance with the results from our prior study [26] and was named "Conscious antibiotic prescription in dentistry." This VLE contained dynamic and interactive multimedia learning resources and a storytelling technique with an animated character represented by a bacterium acting as a moderator, which intended to dominate the world and challenged the participant through different learning moments (Fig 1).

Different types of learning resources were used in the virtual learning environment for conscious antibiotic prescription in dentistry. Reprinted from [VLE "Prescripción consciente de Antibióticos en odontología"] under a CC BY license, with permission from [Universidad Cooperativa de Colombia], original copyright [2020].

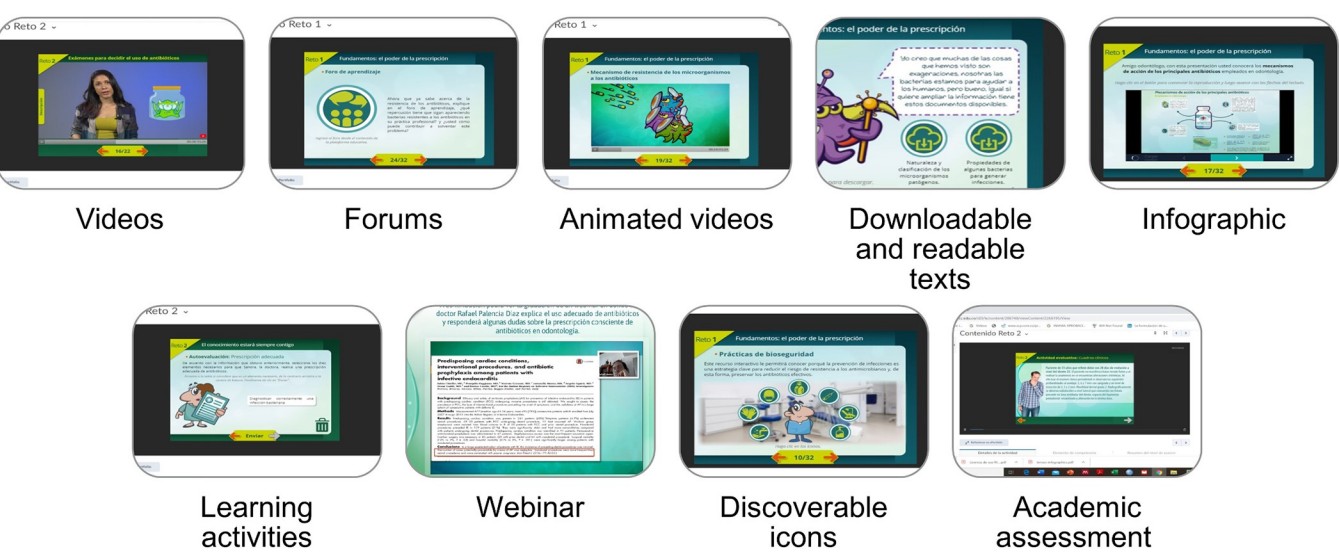

**Fig 1. Multimedia learning resources.**

The Brightspace platform (Desire2Learn, Kitchener, ON, Canada) was used to implement the course; three groups were created, each supported by a collaborator that served as guidance for technical issues and an expert professor with whom the participants interacted during the two learning forums. The course comprised five sections, including three interactive learning challenges. Virtual spaces contained subjects and activities for competence accomplishment, as well as academic assessments (Table 1). The course duration was 8 weeks, with an hourly intensity of 6 h/week. Additionally, a Question & Answer webinar was presented during Challenge 2 by an expert in pharmacology.

## Impact of the VLE on dentists

To determine the effect of the VLE on dentists, participants answered a questionnaire intended to measure their levels of awareness, attitudes, and intention to practice antibiotic prescription before, immediately after, and 6 months post-intervention. The immediate impact ("immediately after" intervention) was compared to the first measurement ("before" intervention) of the total sample of dentists who completed the course. To analyze information retention, results from dentists who answered the questionnaire at 6 months ("6 months" post-intervention) were compared to the data obtained from the same population that answered the initial questionnaire ("before" intervention). The questionnaire was previously validated through a focus group, an expert panel, a pilot test, and a survey of dentists (n = 98) to determine confidence, psychometric index, and one-dimensionality questions (biserial correlation > 0.0, discrimination index > 0, no response index = 0 − 0.15, one-dimensionality index p = 0.930, and internal consistency = 0.810) [27].

The questionnaire consisted of five sections: the first part contained general information questions such as sex, type of practice, years of practice, and specialization; the second segment comprised six questions on "awareness," including antibiotic effectiveness, antibiotic resistance, and the association between prescription and resistance; the third section had eight questions on "attitudes," including decision making on prescription and non-clinical factors influencing prescriptions; the fourth segment comprised 22 questions on "intention to practice" according to the AHA and ADA guidelines, and recent studies [6−8], including clinical

**Table 1. Subjects included in the course sections.**

| Sections | Lerning moment | Subjects |
|---|---|---|
| **Introduction** | Introduction | • Questionnaire regarding awareness, attitudes, and intention to practice antibiotic prescription in dentistry.<br>• Antibiotic resistance problem.<br>• Real situations of patients with resistant bacterial infections. |
| **Challenge 1** | To associate the antibiotic effect on pathogen bacteria and resistance mechanisms they develop. | • Nature and classification of pathogen microorganisms.<br>• Characteristics of bacteria to generate infections.<br>• Principles and practices on infection control.<br>• Mechanisms of action of main antibiotics used in dentistry.<br>• Resistance mechanisms to antimicrobials.<br>• Knowledge of the activity spectrum of common-prescribed antibiotics in dentistry.<br>• Pharmacokinetics and pharmacodynamics of antibiotics.<br>• In challenge 1, the participant had to associate a type of antibiotic with a bacterial infection. |
| **Challenge 2** | To comprehend the clinical picture of patients attending a dental outpatient appointment that results in the appropriate administration of antibiotics according to clinical protocols. | • Key elements for antibiotic prescription in dentistry.<br>• Appropriate use of antibiotics for prophylaxis therapy.<br>• Type of infections (pathologies) that require an antibiotic prescription.<br>• Clinical cases that may or may not require an antibiotic prescription.<br>• Complementary tests for decision making on antibiotic prescription.<br>• Responsibility and legal aspects for an antibiotic prescription.<br>• Leaflet "Do You Need Antibiotics from Your Dentist?" from Centers for Disease Control and Prevention [39].<br>• Challenge 2 contained clinical cases where the dentist had to select whether antibiotic treatment should or should not be used (feedback included). |
| **Challenge 3** | To apply microbiological and clinical foundations for appropriate antibiotic prescription according to a comprehensive health assessment. | • Guidelines and protocols for antibiotic stewardship in dentistry (American Heart Association [AHA] [13], American Association of Endodontists [AAE] [14], American Academy of Pediatric Dentistry [AAPD] [40], American Academy of Orthopedic Surgeons [AAOS]-American Dental Association [ADA]) [41].<br>• Prevention of unnecessary administration of broad-spectrum antibiotics.<br>• Dosage calculation of antibiotics in children.<br>• Clinical cases for antibiotic prescription in dentistry.<br>• Challenge 3 consisted of clinical cases where the participant had to select the best treatment (feedback included). |
| **Course completion** | Final | • Questionnaire regarding awareness, attitudes, and intention to practice antibiotic prescription in dentistry.<br>• Satisfaction and applicability questionnaire. |

cases in which antibiotics are prescribed, as well as the duration and frequency of prescription in the dental practice; and the fifth contained four questions on complementary information. Each question from the sections covering awareness, attitudes, and intention to practice were scored and classified according to the number of correct answers as follows: low level (awareness 0 − 3, attitudes 0 − 4, and intention to practice 0 − 9), medium level (awareness 4 − 5, attitudes 5 − 7, and intention to practice 10 − 13), and high level (awareness 6, attitudes 8, and intention to practice 14 − 22).

## Course satisfaction and applicability

After the course completion, a satisfaction and applicability questionnaire was implemented within the same platform, consisting of pre-coded closed-ended questions and a five-point Likert scale ranging from strongly agree (5 points) to strongly disagree (1 point). Additionally,

questions regarding whether the participant had previously attended virtual courses and the time of dedication to the current course activities were included. Finally, scores from the three academic assessments (one per challenge) were extracted from the platform.

## Statistical analysis

Data analysis was performed using IBM SPSS version 25.0 (IBM, Armonk, NY, USA), in which sociodemographic data were analyzed and frequencies of awareness, attitudes, and intention to practice (individual questions and level classification) were obtained. With the number of correct answers, a descriptive analysis was performed to determine the central tendency (median) and measures of position (quartiles). To assess the effect of the VLE before and immediately after the intervention, the results from participants who completed learning activities, academic assessments, and answered the questionnaire were compared. To evaluate the 6-month impact results from participants who answered the questionnaire at the beginning and after 6 months were compared. Comparative analysis of the number of correct answers (quantitative variable) was performed using the non-parametric Wilcoxon signed-rank test, as the data did not follow a normal distribution. McNemar's test was used to compare the levels of awareness, attitudes, and intention to practice (qualitative variables). A significance level of 5% was used for all statistical tests.

## Results

### Sociodemographic characteristics

A total of 279 dentists initiated the VLE, of which 73.8% completed the course. Of the participants, 69.9% were women, 60.2% had a private clinical practice, 45.6% had more than 10 years of clinical experience, and 27.2% were based in Villavicencio; dentists with or without a postgraduate degree were equal (Table 2). Furthermore, 57.3% and 80.1% of dentists showed a medium level of awareness and attitudes, respectively, and 91.3% had a high level of intention to practice (Table 3).

### Impact of the VLE on awareness

The immediate impact of the VLE among 206 participants showed a statistically significant increase in awareness, specifically in the median of correct answers (before: 5.0 interquartile range [IQR] [4.0 − 6.0], after: 6.0 IQR [5.0 − 6.0], p < 0.001) as well as in the number of participants with a high level of awareness (p < 0.001) (Figs 2A and 3A). Likewise, a significant reduction in the number of participants with low (p < 0.001) and medium levels of awareness (p = 0.001) was observed (Fig 3A). Furthermore, regarding questionnaire, a significant increase in the number of participants who correctly answered most of the questions was detected, including those related to antibiotic efficacy, antibiotic resistance, and the association between antibiotic prescription and the generation of resistance (S1 Appendix).

The 6-month impact among 155 dentists who answered the questionnaire once again showed a significant increase in the median of correct answers (before: 5.0 IQR [4.0 − 5.25], after: 5.0 IQR [4.0 − 6.0], p < 0.001) (Fig 2A) and in the number of participants who correctly answered the questions from this section, except in the definition of antibiotic resistance (S1 Appendix). Similarly, an increase in the number of participants who showed a high level of awareness was observed; however, this result did not show a statistically significant difference (p = 0.161). Finally, the number of participants with a medium level was similar (p = 1.000), whereas a statistically significant reduction in those with a low level of awareness was obtained (p = 0.019) (Fig 3A).

**Table 2. Sociodemographic characteristics of the participants (n = 206).**

| Characteristic | Absolute Number | Percentage (%) |
|---|---|---|
| **Sex** | | |
| Women | 144 | 69.9 |
| Men | 62 | 30.1 |
| **City** | | |
| Barranquilla | 25 | 12.1 |
| Bogota | 34 | 16.5 |
| Cali | 10 | 4.9 |
| Cartagena | 34 | 16.5 |
| Medellin | 16 | 7.8 |
| Monteria | 31 | 15 |
| Villavicencio | 56 | 27.2 |
| **Type of practice** | | |
| Private | 124 | 60.2 |
| Public | 33 | 16 |
| Mixed | 49 | 23.8 |
| **Years of practice** | | |
| <5 | 81 | 39.3 |
| 6–10 | 31 | 15 |
| >10 | 94 | 45.6 |
| Dental Specialization | 103 | 50 |
| **Specialized dentists** | | |
| Endodontics | 16 | 15.5 |
| Health Administration/Management | 16 | 15.5 |
| Periodontics | 15 | 14.6 |
| Pediatric Dentistry/Pediatric Stomatology | 11 | 10.7 |
| Oral/Maxillofacial Surgery | 10 | 9.7 |
| Oral Rehabilitation/Implantology | 7 | 6.8 |
| Health Services Audit | 5 | 4.9 |
| Orthodontics | 4 | 3.9 |
| Basic Sciences | 4 | 3.9 |
| Orthopedics | 3 | 2.9 |
| Health Promotion | 2 | 1.9 |
| Education | 2 | 1.9 |
| Esthetics | 1 | 1.0 |
| Prosthodontics | 1 | 1.0 |
| Other | 6 | 5.9 |

## Impact of the VLE on attitudes

A significant increase in the median of correct answers immediately after the course completion was observed (before: 6.0 IQR [5.0 − 7.0], after: 6.0 IQR [6.0 − 7.0], p < 0.001) (Fig 2B), and in the number of participants who correctly answered two questions, which were related to prescription based on patients' symptoms and prescription following international guidelines (S1 Appendix). Moreover, a significant reduction in the number of participants with a low level (p = 0.001) and a non-significant increase in the medium (p = 0.152) and high levels of attitudes (p = 0.230) was observed (Fig 3B).

Regarding the 6-month impact, the median of correct answers was slightly lower and did not reach statistical significance (before: 6.0 IQR [5.0 − 7.0], after: 6.0 IQR [5.0 − 6.0],

**Table 3. Baseline levels of awareness, attitudes, and intention to practice of the participants (n = 206).**

| Characteristics | Absolute Number | Percentage (%) |
|---|---|---|
| **Levels of awareness** | | |
| Low | 28 | 13.6 |
| Medium | 118 | 57.3 |
| High | 60 | 29.1 |
| **Levels of attitudes** | | |
| Low | 30 | 14.6 |
| Medium | 165 | 80.1 |
| High | 11 | 5.3 |
| **Levels of intention to practice** | | |
| Low | 3 | 1.5 |
| Medium | 15 | 7.3 |
| High | 188 | 91.3 |

p = 0.105) (Fig 2B). Similarly, a non-significant increase in the medium level of attitudes (p = 0.392) and a non-significant reduction in the high level (p = 0.092) was detected (Fig 3B). Finally, the correct answer to one question from this section (prescription following international guidelines) remained significantly high (p = 0.014) (S1 Appendix).

## Impact of the VLE on intention to practice

In general, a statistically significant increase in the median of correct answers was observed (before: 17.0 IQR [15.0 − 18.0], after: 17.0 IQR [16.0 − 18.0], p = 0.042) (Fig 2C). Regarding levels of intention to practice, a reduction in the number of participants with low (p = 0.500) and medium levels (p = 0.523) and an increase at the high level (p = 0.286) was evident; all differences were not statistically significant (Fig 3C). A favorable effect of the VLE was detected, as most of the participants considered it uncommon to prescribe antibiotics in their dental practice (p < 0.001) and would not prescribe antibiotics for acute gingivitis and stomatitis (p = 0.036), after simple tooth extraction (p < 0.001), endodontic procedures (p < 0.001), and implant placement (p < 0.001) (S1 Appendix). In the question regarding antibiotic prophylaxis to prevent IE, a positive effect was observed with the correct answer on antibiotic prescription in patients with a pacemaker (p < 0.001). Conversely, a significant reduction in the number of participants who correctly answered the question concerning patients with autoimmune diseases (p = 0.005), under immunosuppressive therapy (p = 0.037), and diagnosed with AIDS (p = 0.017) was observed (S1 Appendix). The above findings reflect confusion among dentists regarding antibiotic prophylaxis.

The 6-month impact showed a significant increase in the median of correct answers (before: 17.0 IQR [15.0 − 18.0], after: 18.0 IQR [16.0 − 19.0], p < 0.001) (Fig 2C), and in the number of participants who correctly answered most of the questions from this section, which demonstrates that intention to practice antibiotic prescription is maintained over time (p < 0.01, p < 0.05) (S1 Appendix). Unlike immediate impact, an increase in the number of participants who correctly answered the question regarding antibiotic prophylaxis to prevent IE in patients with autoimmune diseases, those undergoing immunosuppressive therapy, and those diagnosed with AIDS was obtained; however, this was not statistically significant (S1 Appendix). Regarding levels of intention to practice, none of the participants were at a low level, there was a slight increase at the medium level (p = 0.804), and a high level was maintained over time (p = 1.000) (Fig 3C).

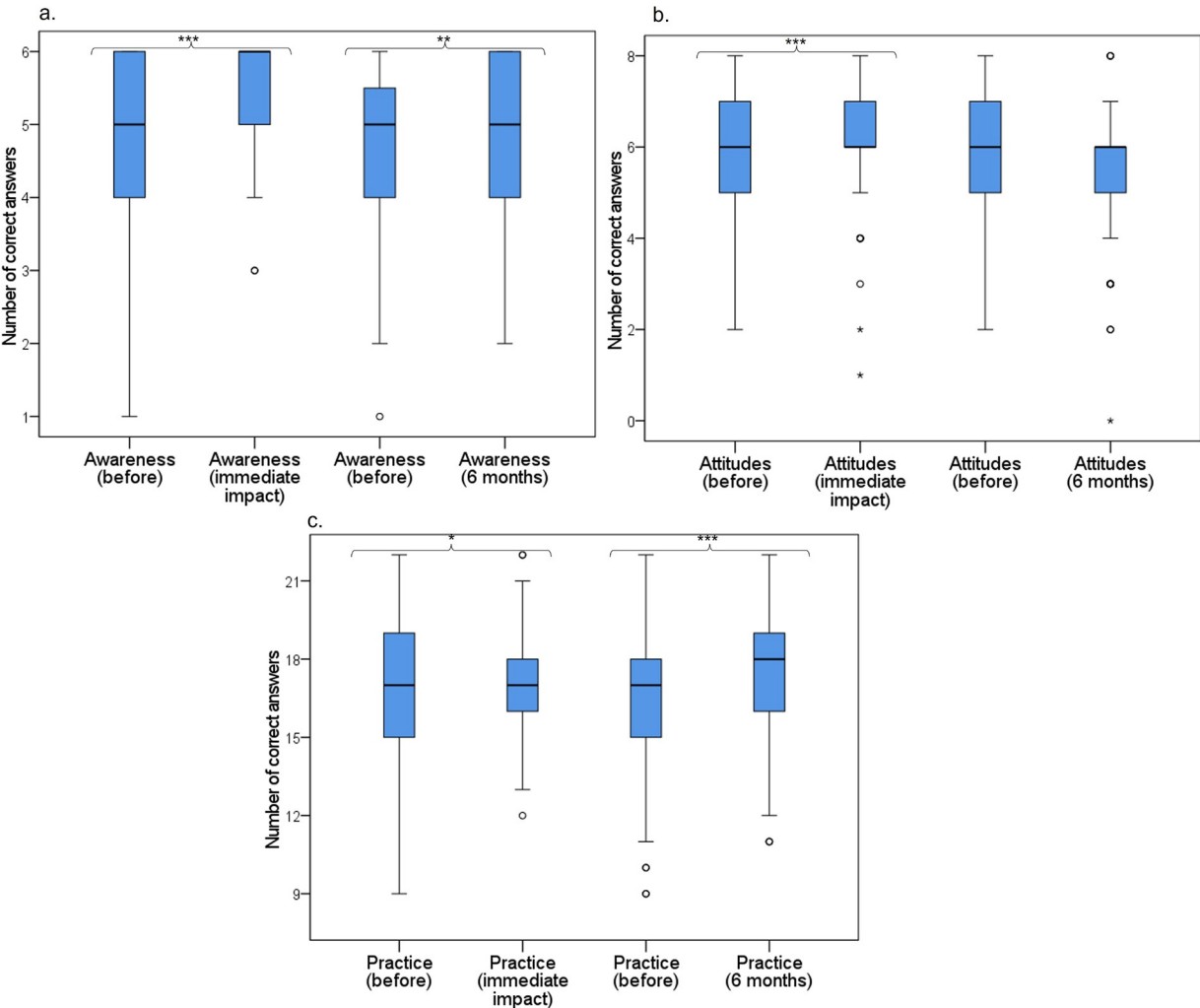

**Fig 2. Impact of the virtual learning environment on the number of correct answers.** The number of correct answers on awareness (a), attitudes (b), and intention to practice (c) before the application of the questionnaire, immediately after, and 6 months post-intervention. The median and interquartile ranges are shown. Differences are determined using Wilcoxon signed-rank test. *p < 0.05, **p < 0.01, ***p < 0.001.

### Immediate and 6-month impact of the VLE on other questions

The week before the application of the first questionnaire, patients receiving antibiotics were reduced as a high number of participants did not prescribe antibiotics to any of their patients (p < 0.001) or prescribed to 1–5 patients (p < 0.001), 6–10 patients (p = 0.388), and > 10 patients (p < 0.039) (Fig 4A). Moreover, these results were obtained during the coronavirus disease (COVID-19) lockdown and should be interpreted with caution.

At 6 months, the number of dentists not prescribing antibiotics to any patient increased; however, this was not statistically significant (p = 0.392). Conversely, a slight reduction in the number of participants who prescribed antibiotics to 1–5 patients was observed (p = 0.085); however, the number of dentists who prescribed to 6–10 patients (p = 1.000) and > 10 patients remained the same (p = 1.000) (Fig 4A). Among the first-choice antibiotics, amoxicillin was preferred by dentists, followed by amoxicillin/clavulanic acid. After implementing the VLE, a slight inclination for azithromycin, metronidazole, clindamycin, and cephalexin was detected (Fig 4B).

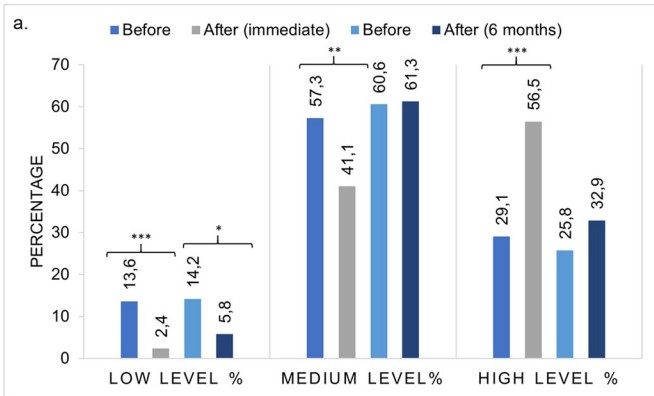
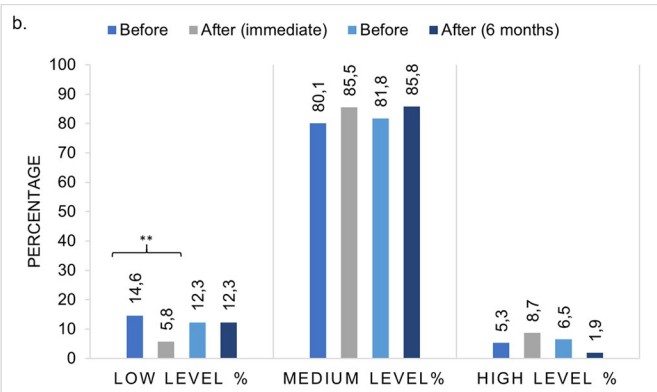
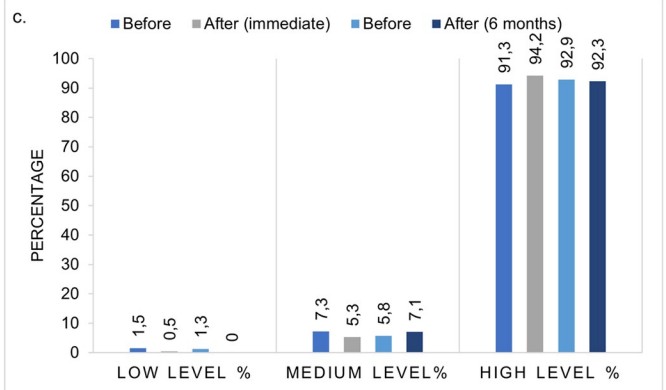

**Fig 3. Impact of the virtual learning environment on levels.** Percentage of participants showing low, medium, or high levels of awareness (a), attitudes (b), and intention to practice (c) before the application of the questionnaire, immediately after, and 6 months post-intervention. The median is shown. Differences are determined using McNemar's test. $^{*}p < 0.05$, $^{**}p < 0.01$, $^{***}p < 0.001$.

## Academic assessments, level of satisfaction, and course applicability

The median score of academic assessments from participants that completed the course was 4.4 IQR [3.6 – 4.8], and the median in each learning moment was 5.0 IQR [2.5 – 5.0] for Challenge 1, 5.0 IQR [4.2 – 5.0] for Challenge 2 and 4.5 IQR [3.9 – 5.0] for Challenge 3 (Fig 5A). Of the participants, 60.5% had previously completed a virtual course, 82.2% did not have any issue understanding the sections of the VLE, and 56.8% reported a dedication of time between 3 and 5 hours per week (Fig 5B).

Generally, participants strongly agreed with most of the questions related to the quality of the course (77.9% – 88.5%). Regarding applicability, the question associated with the acquisition of consciousness on antibiotic prescription was the most strongly agreed by professionals (92.1%), followed by application of knowledge (91%) and the intention to modify prescription (84.2%) (Fig 5C).

## Discussion

The findings from this study allowed us to reject the null hypothesis, as significant differences in awareness and intention to the practice of participants on antibiotic prescription were observed. Furthermore, the VLE developed and implemented in the present study confirmed the strengths of virtual education, such as flexibility, affordability, accessibility, and overcoming of geographic and time limitations [17, 20], favoring the formation process of the conscious prescription of antibiotics on dentists from seven Colombian cities. The results showed

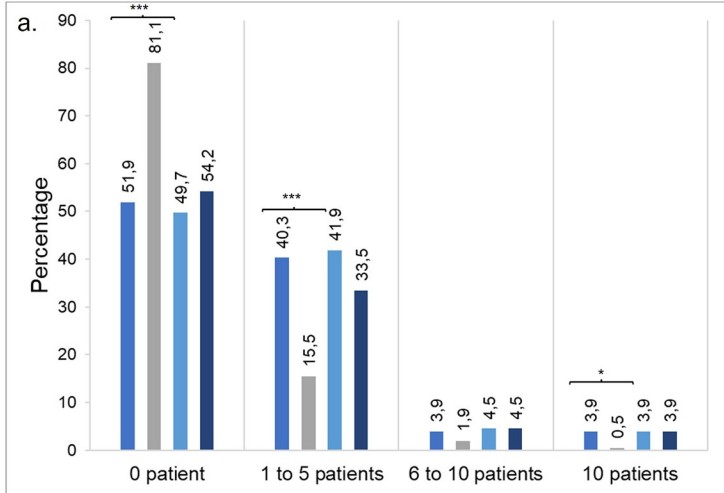

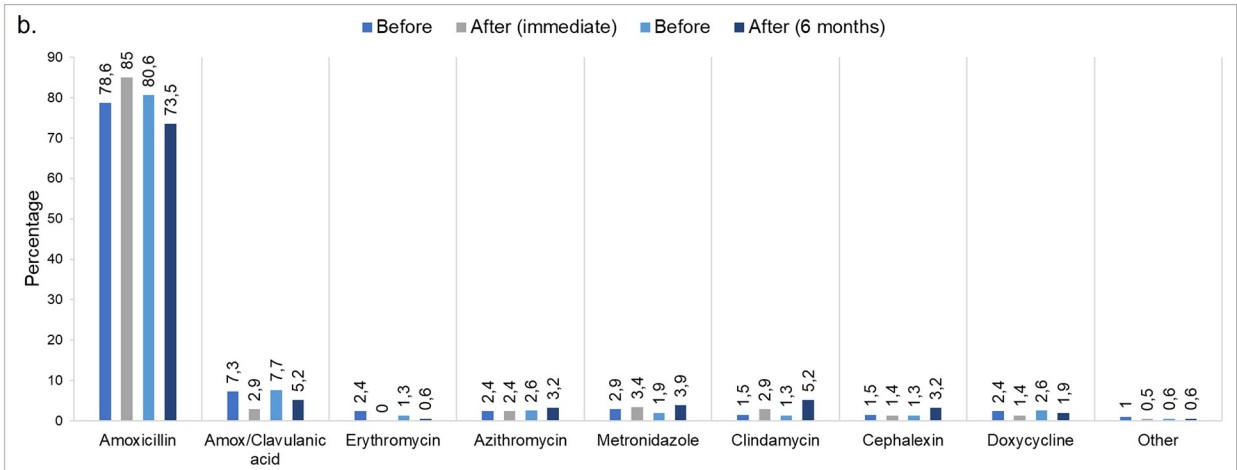

**Fig 4. Impact of the virtual learning environment on antibiotic prescription.** (a) Percentage of participants that prescribed antibiotics the week before the application of the questionnaire, immediately after, and 6 months post-intervention. (b) First choice antibiotic among participants. Differences are determined using McNemar's test. *p < 0.05, **p < 0.01, ***p < 0.001.

that implementation of the VLE had an immediate positive effect on the median of correct answers and an increase in the number of dentists that correctly answered some of the questions from the questionnaire. Additionally, an increase in the number of dentists with improved levels of consciousness was observed, particularly their levels of awareness. The effectiveness of virtual learning spaces on the acquisition of knowledge and abilities for antibiotic prescription has also been demonstrated in different medical disciplines [28, 29] and dentistry [19, 30], which confirms the relevance of virtual scenarios in the transfer of knowledge on healthcare careers.

Regarding the 6-month impact, a significant increase in the median of correct answers on awareness and intention to practice as well as the number of dentists that correctly answered some of the questions from the questionnaire was maintained; however, information retention was not compared to a control group. Unlike this study, other authors implemented a VLE on antibiotic prescription in medical students and found significantly higher results when compared to a control group (non-virtual education) [28].

Additionally, at 6 months, the present study identified a reduction in the median number of correct answers and the number of dentists that reached a high level of awareness compared

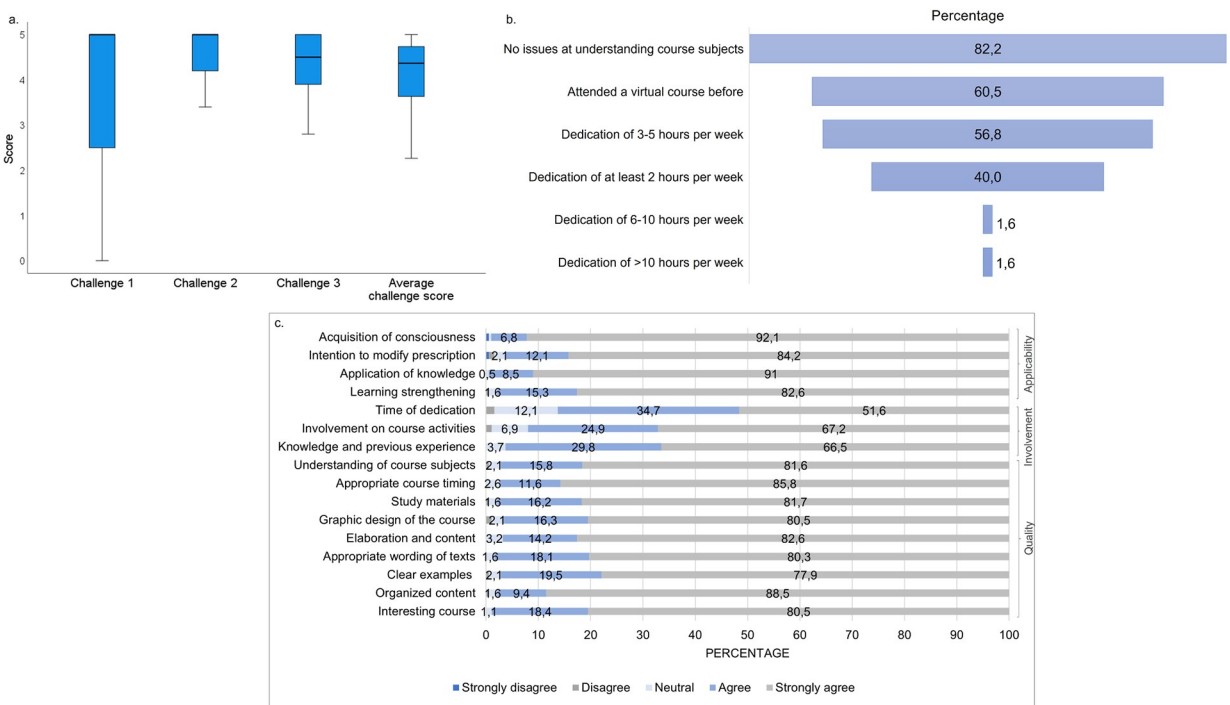

**Fig 5. Academic assessments, level of satisfaction, and applicability of the virtual learning environment.** (a) Score of academic assessments from Challenges 1, 2, and 3. Median and interquartile ranges are shown. The maximum score is 5.0. (b) Level of difficulty and time of dedication to the course. (c) Perception of participants on applicability, involvement, and quality of the course.

to the immediate impact. In another study that implemented virtual tools, despite participants obtaining high scores immediately after the intervention, a non-significant reduction in theoretical test scores after 2 months was observed [29]. For this reason, the combination of multifaceted strategies that include audit trails and feedback processes to achieve a significant impact on antibiotic prescriptions is recommended [17].

Regarding attitudes toward antibiotic prescription 6 months after the course, the results showed that dentists could not modify them. This may be because attitudes are stable entities stored in the memory, implying that new information competes with previously-stored judgments [14]. Therefore, the above suggests that a more robust strategy for individual attitudes must be considered before implementing virtual environments.

Antibiotic prophylaxis is one of the primary actions that should be improved in dental practice, as antibiotics are commonly prescribed before dental procedures to reduce bacteremia [31]. However, some participants could not distinguish antibiotic prophylaxis to prevent IE and complications in patients with certain systemic conditions, which may be due to insufficient time spent on course activities and might have negatively impacted the understanding of antibiotic prophylaxis. Additionally, confusion between prophylactic and therapeutic prescriptions among dentists has been highlighted in the literature [32, 33] and requires a robust educational strategy to overcome the current situation in our field.

Due to the COVID-19 pandemic, the immediate impact of the VLE on the frequency of antibiotic prescriptions could not be explained. Interestingly, after 6 months, dentists resumed their clinical practice and prescribed fewer antibiotics to their patients, although whether the COVID-19 pandemic affected dental outpatient appointments remains undetermined. Accordingly, a study showed that implementing a multimodal strategy comprising an

educative section and an online tool on antibiotic prescription in dentistry demonstrated a reduction in antibiotic prescription and inappropriate indications in the long term [19].

Amoxicillin remains the first-choice antibiotic among prescribed antibiotics; however, other antibiotics are being considered among dentists. This may be explained by the fact that amoxicillin was included in several clinical cases during this course, as this antibiotic has demonstrated safety and efficacy in dentistry [34]; therefore, it is the first choice by most dentists, as described elsewhere [35]. Nevertheless, it would be interesting to evaluate the use of other types of antibiotics in a broader context that can be used safely in dental practice. For example, the Australian Therapeutic Guidelines recommend phenoxymethylpenicillin prescription against oral bacteria (primarily gram-positive) due to its high efficacy (85%) and reduced spectrum compared with amoxicillin [19, 32].

Regarding scores obtained during the academic activities, good performance of most participants on academic assessments was observed, with the assessments from Challenge 1 (association of a type of antibiotic with a bacterial infection) and Challenge 2 (clinical cases to select whether antibiotics should or should not be prescribed) producing better scores than those obtained from Challenge 3 (clinical cases to select the best treatment). In a similar study, an online course given to 310 medical students showed that the majority had an average score of 90% on activities related to basic knowledge (principles of antibiotic stewardship). Moreover, 153 of these participants completed all five interactive online clinical cases, with approximately half scoring over 90% in the first attempt [36]. Therefore, the transfer of theoretical knowledge through virtual tools is more effective than its integration, which represents an important challenge for the development of these types of environments and encourages the exploration of alternative technologies such as virtual reality [37, 38].

Finally, the level of satisfaction and course applicability among participants was high, indicating good acceptance of the virtual tool. It has been reported that virtual education is well accepted compared to traditional education as it promotes an attractive, interactive, and cognitive formation process through multimedia resources that facilitate learning [24, 38]. Thus, the acceptance of our virtual tool is superior to that of other virtual strategies [36, 39]. Although the impact of the VLE on attitudes was not significant, this course may contribute to increasing awareness of the appropriate use of antibiotics in Colombia and other Hispanic countries. In addition, it might have an impact on bacterial resistance and in the reduction of adverse effects such as diarrhea, allergic reactions, anaphylaxis, and infection by *Clostridium difficile* [40]. Therefore, this course was provided on an open-access platform to reach more dentists.

A limitation of the present study is related to the absence of a control group receiving a traditional strategy or non-virtual education to compare the impact of the VLE, which was due to the distance between the seven Colombian cities and the interest in implementing the VLE to a higher number of dentists. Similarly, score categorization was a limitation, as according to the percentile classification during the questionnaire validation, a high level of awareness and attitudes was achieved with 100% of correct answers, whereas for intention to practice, a high level was reached with 63 – 100% of correct answers [27]. Finally, other limitations include a selection bias as dentists had participated in our previous study where the questionnaire to determine awareness, attitudes, and intention to practice antibiotic prescription was implemented [26], and the COVID-19 pandemic and the lockdown in Colombia from March to August 2020, which did not allow us to measure the frequency of antibiotic prescription among participants and, consequently, the impact of the VLE on practices.

Despite this, new knowledge, reinforcement of the concepts regarding antibiotic prescription, acquisition of consciousness, and intention to modify prescription resulted from the VLE implemented here. These are the starting points that may contribute to preventing the generation of antibiotic-resistant bacteria [41]. However, further interactive and reflexive activities

that improve attitudes, a deep understanding of the indications for antibiotic prophylaxis, and knowledge retention should be reinforced.

## Conclusions

This study successfully developed and implemented a VLE on conscious antibiotic prescription in dentistry, with dynamic and interactive learning that contributed to improving Colombian dentists' awareness and intention to practice. However, it is important to highlight that low information retention on attitudes after 6 months demonstrates the need to develop a more robust approach that modifies this factor and effectively contributes to promoting the appropriate use of antibiotics.

## Supporting information

**S1 Appendix. Immediate and 6-month impact of the virtual learning environment (VLE) implementation.** Impact on the number of participants who correctly answered questions from the questionnaire.
(DOCX)

## Acknowledgments

We thank Camilo Toscano and Cristian Metaute for their contribution during the development of the virtual learning environment and all participants for their commitment to this study.

## Author Contributions

**Conceptualization:** María del Pilar Angarita-Díaz, Diana Forero-Escobar, Angélica Ricaurte-Avendaño, Julián Mora-Reina.

**Formal analysis:** María del Pilar Angarita-Díaz.

**Investigation:** María del Pilar Angarita-Díaz, Lilia Bernal-Cepeda, Leidy Bastidas-Legarda, Diana Forero-Escobar, Angélica Ricaurte-Avendaño, Julián Mora-Reina, Martha Vergara-Mercado, Alejandra Herrera-Herrera, Martha Rodriguez-Paz, Sandra Cáceres- Matta, Natalia Fortich-Mesa, Emilia María Ochoa-Acosta.

**Methodology:** María del Pilar Angarita-Díaz, Lilia Bernal-Cepeda, Diana Forero-Escobar, Angélica Ricaurte-Avendaño, Julián Mora-Reina, Martha Vergara-Mercado, Alejandra Herrera-Herrera, Martha Rodriguez-Paz, Sandra Cáceres- Matta, Natalia Fortich-Mesa, Emilia María Ochoa-Acosta.

**Project administration:** María del Pilar Angarita-Díaz.

**Software:** Angélica Ricaurte-Avendaño.

**Supervision:** María del Pilar Angarita-Díaz, Diana Forero-Escobar.

**Writing – original draft:** María del Pilar Angarita-Díaz, Leidy Bastidas-Legarda.

**Writing – review & editing:** Lilia Bernal-Cepeda, Leidy Bastidas-Legarda.

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
