## [Decision Letter · Decision Letter 0]

14 May 2021

PONE-D-21-04062

Impact of a Virtual Learning Environment on the conscious prescription of antibiotics in Colombian dentists

PLOS ONE

Dear Dr. Angarita Díaz,

Thank you for submitting your manuscript to PLOS ONE. After careful consideration, we feel that it has merit but does not fully meet PLOS ONE’s publication criteria as it currently stands. Therefore, we invite you to submit a revised version of the manuscript that addresses the points raised during the review process.

We look forward to receiving your revised manuscript.

Kind regards,

Prof. Dr. med. dent. Dr. h. c. Andrej M Kielbassa

Academic Editor

PLOS ONE

Journal Requirements:

2. We note that Figure 1 in your submission contain copyrighted images. All PLOS content is published under the Creative Commons Attribution License (CC BY 4.0), which means that the manuscript, images, and Supporting Information files will be freely available online, and any third party is permitted to access, download, copy, distribute, and use these materials in any way, even commercially, with proper attribution. For more information, see our copyright guidelines: http://journals.plos.org/plosone/s/licenses-and-copyright.

2.1.    You may seek permission from the original copyright holder of Figure 1 to publish the content specifically under the CC BY 4.0 license.

2.2.    If you are unable to obtain permission from the original copyright holder to publish these figures under the CC BY 4.0 license or if the copyright holder’s requirements are incompatible with the CC BY 4.0 license, please either i) remove the figure or ii) supply a replacement figure that complies with the CC BY 4.0 license. Please check copyright information on all replacement figures and update the figure caption with source information. If applicable, please specify in the figure caption text when a figure is similar but not identical to the original image and is therefore for illustrative purposes only.

3. We note that Figure 1 includes an image of a participant in the study. 

Reviewers' comments:

Reviewer's Responses to Questions

**Comments to the Author**

1. Is the manuscript technically sound, and do the data support the conclusions?

Reviewer #1: Yes

Reviewer #2: Yes

Reviewer #3: Yes

2. Has the statistical analysis been performed appropriately and rigorously? 

Reviewer #1: Yes

Reviewer #2: Yes

Reviewer #3: I Don't Know

3. Have the authors made all data underlying the findings in their manuscript fully available?

Reviewer #1: Yes

Reviewer #2: Yes

Reviewer #3: Yes

4. Is the manuscript presented in an intelligible fashion and written in standard English?

Reviewer #1: Yes

Reviewer #2: Yes

Reviewer #3: No

5. Review Comments to the Author

Reviewer #1: The article was well written, and the study was well designed. However, some doubts remained:

1. How were the participants included in each city? Was the sample probabilistic or non-probabilistic? It is necessary to describe more details;

2. The authors must discuss the absence of a control group;

3. The objectives and the conclusions must be aligned; the authors did not cite the development of a Virtual Learning Environment in the objectives;

4. Why did dentists were unable to modify their attitudes on antibiotic prescription six months after the course, besides modifying their awareness? Due complexity of this theme, maybe is necessary a more profound discussion based on educational psychology. Courses cannot be the best strategy.

Reviewer #2: General remark

- Please revise for minor typos, and double check grammar. See, for example, "however this was not" and "however, the number of". Revise for missing commas.

Abstract

- Please see Guidelines for Authors, and consult some recently published Plos One papers. No subheadings mandatory here. Please delete "Background:", "Methods :", "Results:", and "Conclusion :". Revise carefully for proper spacebar use.

- Please provide complete results, and add exact P values on a 3-digit basis (example: p=0.109). Revise carefully.

- Remember that this section allows for 300 words, see Guidelines. Currently, your word count would be 216, and it seems unclear why you obviously have failed to provide a complete section presenting the most important outcome convincingly.

- With your conclusions, please do not simply repeat your results. Instead, stick to your aims, and provide a sound and reasonable extension of your outcome. This extension must be based on your results, but a simple repetition would not seem convincing.

- This section has been poorly elaborated only. There are 12 (!) (co-)authors of this draft, all of them having read and approved this manuscript, right? This should result in a flawless paper, without any typos, and with sound conclusions. I hope you will agree.

Intro

- Would seem basically sound.

- Please provide your objectives. Remember that there is a clear difference between "aims" and "objectives".

- A sound null hypothesis is missing. Note that H0 must be deducible from the forgoing thoughts.

Meths

- Do not use legal terms like "®", and so on. This is unusual with scientific writing, please delete.

- - With ALL materials and methodologies (including statistical software), please use general names with your text, followed by (brand name; manufacturer, city, STATE (abbreviated, if US), country) in parentheses. Stick to semicolon. Revise thoroughly.

Results

- Would seem sound.

Disc

- Stick to H0 when staring this section.

- Do not use authors' names with your text. The latter will be acknowledged with your Reference list. Delete "Sikkens et al.", and focus on your main thoughts.

- Same with "Soltanimehr et al", and with "Teoh et al.,".

- Please add some educated thoughts on the generalizability of your outcome, to Columbia, and to other countries.

Concl

- Your aims were "(...) this study aimed to determine the impact of a VLE designed to improve awareness, attitudes and intention to practice on antibiotic prescription". With your conclusions, please stick exclusively to your aims, and provide a reasonable extension of your outcome.

- "High scores on course characteristics and applicability such as design, content, quality, organization, timing as well as in the acquisition of knowledge, acquisition of consciousness and intention to modify prescription were received." This might be right, but it does not stick to your aims. Revise carefully.

Refs

- Please see recent Plos One papers, and add pmid numbers.

All in all, the authors have submitted a nice draft, considered interesting and easily intelligible. This paper should be worth following after major revisions.

Reviewer #3: Thank you for the opportunity to review your paper. There is clearly much work which has been undertaken and your findings are an important addition to the literature. However, the very comprehensive nature of your paper makes it rather difficult to read.

General - explaining some of the technical terminology would be helpful - I do not understand 'intention to practice' - does this mean intention to prescribe antibiotics within the US guidelines which you have highlighted as the basis for Colombian dentistry?

Introduction - Too many references - 37 in just 5 paragraphs. Please think really carefully about the best one or two to chose to make each point rather than including up to 5 citations per sentence. For example, your first sentence should include the FDI World Dental Federation's policy statement or white paper on tackling overprescribing of antibiotics in dentistry ahead of many of the other references. And in the opening sentence of your third paragraph, could I suggest a single systematic review which looked at why dentists prescribe antibiotics unnecessarily: https://academic.oup.com/jac/article/74/8/2139/5475276

Introduction - para 2 - guidelines for appropriate dental antibiotic prescribing differ around the world so you need to be clear in this paper that you are referring to that which is deemed appropriate in Colombia. And I think you have a typo as periodontal abscesses are less likely to be life-threatening than periapical abscesses. Later in the same paragraph, you state that orthopaedic surgeons 'must' define the antibiotic regimen - this does not make sense given the first part of this sentence.

Introduction - para 3 - other than the first sentence of this paragraph, much of it would sit more comfortably in the discussion section.

Methods - Be clear about when you undertook this study - this is particularly important as you refer to it in the discussion section.

Thinning out the detail to that which is essential will make this section much clearer. Don't repeat information which is already in the tables. And be careful of jargon: alpha risk and beta risk mean nothing to the general reader. Simply say that your sample size was calculated as 140 dentists based on a cut off for significance of p=0.05 and .... Also, could you mention the date when the ethics committee approved it?

Table 1 - this needs to be streamlined to get rid of superfluous detail (eg 'welcome message' and 'course detail')

Results - Again don't repeat information from the tables in the text. For example in the second sentence just say 'Details of the sociodemographic characteristics of participating dentists are included in Table 2.' In spite of its title, you have included more than just sociodemographic characteristics in Table 2. Baseline levels of awareness, attitudes and 'intention to practice' should be in a separate table. And Table 3 has WAY too much detail for the paper. You need to find a way to summarise this into a format digestible by the reader - or put it into an on-line appendix.

Discussion - the key messages from the paper need to come through this section much more clearly. Given the extensive use of the US guidelines in the rest of the paper, I was rather surprised that you didn't reference any of the US work on dental antibiotic stewardship from Suda, Durkin or Goff.

6. PLOS authors have the option to publish the peer review history of their article (what does this mean?). If published, this will include your full peer review and any attached files.

Reviewer #1: No

Reviewer #2: No

Reviewer #3: No

---

## [Author Response · Author response to Decision Letter 0]

29 Aug 2021

Dear Editor:

Prof. Dr. Andrej M Kielbassa

Academic Editor

PLOS ONE

Please, in the attached file, you will find the revised manuscript PONE-D-21-04062: " Impact of a Virtual Learning Environment on the conscious prescription of antibiotics in Colombian dentists" by Angarita Diaz et al., that we are submitting for publication in "PLOS ONE". We have added responses to the suggestions and comments of the reviewers and would like to thank them for all their time and consideration of our work.

Best regards,

The authors

RESPONSE JOURNAL REQUIREMENTS:

-Comments to the Author

-Please ensure that your manuscript meets PLOS ONE's style requirements, including those for file naming. 

Response: We confirm that our manuscript meets PLOS ONE’s style requirements. Also, the file naming meets the requested style.

-We note that Figure 1 in your submission contain copyrighted images. All PLOS content is published under the Creative Commons Attribution License (CC BY 4.0), which means that the manuscript, images, and Supporting Information files will be freely available online, and any third party is permitted to access, download, copy, distribute, and use these materials in any way, even commercially, with proper attribution.

Response: The images used in this manuscript contain the Creative Commons Attribution License (CC BY 4.0). The course is a product of this research and is Open Access.

We uploaded the completed Content Permission Form in our submission, and we added in the figure caption the following: Reprinted from [VLE “Prescripción consciente de Antibióticos en odontología”] under a CC BY license, with permission from [Universidad Cooperativa de Colombia], original copyright [2020]”. Line 157 (manuscript with track changes).

-We note that Figure 1 includes an image of a participant in the study.

Response: We inform you that Figure 1 does not include an image of a participant in the study. The person who appears is a teacher of the webinar. However, we reduced the size of this photography with the idea of avoiding the image of the person.

-Thank you for including your ethics statement on the online submission form: 

"The present study was approved by the ethics subcommittee from Universidad Cooperativa de Colombia (No. 015-2018). A Written consent was obtained through the VLE.". 

To help ensure that the wording of your manuscript is suitable for publication, would you please also add this statement at the beginning of the Methods section of your manuscript file (to include type of consent provided by participants).

Response: We added the request. Line 137 (track changes).

- Thank you for your response to our copyright queries. Could you please confirm whether the person in figure 1 is shown as part of the “Prescripción consciente de Antibióticos en odontología” under the CC BY 4.0 license? If this identifying image is being republished it will be acceptable to include their image, but otherwise it will need to be removed.

Response: We confirm that the person in figure 1 is part of the “Prescripción consciente de Antibióticos en odontología” under the CC BY 4.0 license.

RESPONSE TO REVIEWERS

REVIEWER 1 

Dear reviewer, thanks for your comments that support us to improve the manuscript. We specify the changes and corrections made as follows:

-Comments to the Author

How were the participants included in each city? Was the sample probabilistic or non-probabilistic? It is necessary to describe more details.

Response: We included specifications about the sample and how we incorporated the participants in each city. Line 143 (manuscript with track changes).

- The authors must discuss the absence of a control group

Response: In the discussion we explained the absence of a control group. Line 384 (track changes).

- The objectives and the conclusions must be aligned; the authors did not cite the development of a Virtual Learning Environment in the objectives.

Response: We modified the objectives and conclusions, including the development of the Virtual Learning Environment in the objectives. Line 131 and line 472 (track changes).

- Why did dentists were unable to modify their attitudes on antibiotic prescription six months after the course, besides modifying their awareness? Due complexity of this theme, maybe is necessary a more profound discussion based on educational psychology. Courses cannot be the best strategy.

Response: In the discussion we explained why dentists may not be able to modify their attitudes on antibiotic prescription 6 months after the intervention. Line 401 (track changes).

RESPONSE TO REVIEWERS

REVIEWER 2 

Dear reviewer, thanks for your comments that support us to improve the manuscript. We specify the changes and corrections made as follows:

Abstract

- Please see Guidelines for Authors,and consult some recently published Plos One papers. No subheadings mandatory here. Please delete "Background:", "Methods:", "Results:", and "Conclusion :". Revise carefully for proper spacebar use.

Response: We modified the abstract, deleting "Background", "Methods", "Results", and "Conclusion" and we revised spacebar use. Line 30 (manuscript with track changes).

- Please provide complete results and add exact P values on a 3-digit basis (example: p=0.109). Revise carefully.

Response: We added the p-value on a 3-digit basis throughout the manuscript. Line 45 (track changes).

- Remember that this section allows for 300 words, see Guidelines. Currently, your word count would be 216, and it seems unclear why you obviously have failed to provide a complete section presenting the most important outcome convincingly.

Response: We completed the abstract presenting the most important outcomes. Line 30 (track changes).

- With your conclusions, please do not simply repeat your results. Instead, stick to your aims, and provide a sound and reasonable extension of your outcome. This extension must be based on your results, but a simple repetition would not seem convincing.

Response: We modified the abstract without repeating the results. We sticked the conclusion to our aims. Line 53 (track changes).

- This section has been poorly elaborated only. There are 12 (!) (co-)authors of this draft, all of them having read and approved this manuscript, right? This should result in a flawless paper, without any typos, and with sound conclusions. I hope you will agree.

Response: Dear reviewer, thanks for your comments. We agree that the abstract needed a more thorough elaboration. We modified this section. Line 30(track changes).

Intro

- Would seem basically sound.

- Please provide your objectives. Remember that there is a clear difference between "aims" and "objectives".

Response: 

We added the objectives. Line 131 (track changes).

- A sound null hypothesis is missing. Note that H0 must be deducible from the forgoing thoughts.

Response: We modified the last section of the introduction with the idea to make the hypothesis deductible. Line 129 (track changes).

Meths

- Do not use legal terms like "®", and so on. This is unusual with scientific writing, please delete.

Response: We deleted the legal terms. Line 164 (track changes).

- With ALL materials and methodologies (including statistical software), please use general names with your text, followed by (brand name; manufacturer, city, STATE (abbreviated, if US), country) in parentheses. Stick to semicolon. Revise thoroughly.

Response: We included brand name information, manufacturer, city, state, and country. Line 164 and line 214 (track changes).

Results

- Would seem sound.

Disc

- Stick to H0 when staring this section.

Response: We modified this section to improve the discussion of the hypothesis. Line 367 and line 371 (track changes).

- Do not use authors' names with your text. The latter will be acknowledged with your Reference list. Delete "Sikkens et al.", and focus on your main thoughts.

Response: We deleted authors' names. Line 387 (track changes).

- Same with "Soltanimehr et al", and with "Teoh et al.,"

Response: We deleted authors' names. Line 394 and line 419 (track changes).

- Please add some educated thoughts on the generalizability of your outcome, to Columbia, and to other countries.

Response: We included information about the generalization of our outcomes to Colombia and other Hispanic countries. Line 453 (track changes).

Conclusion

- Your aims were "(...) this study aimed to determine the impact of a VLE designed to improve awareness, attitudes and intention to practice on antibiotic prescription". With your conclusions, please stick exclusively to your aims, and provide a reasonable extension of your outcome.

Response: We wrote the conclusion having in mind the objective of the study. Line 472 (track changes).

- "High scores on course characteristics and applicability such as design, content, quality, organization, timing as well as in the acquisition of knowledge, acquisition of consciousness and intention to modify prescription were received." This might be right, but it does not stick to your aims. Revise carefully. 

Response: In the conclusion, we deleted information about scores on course characteristics and applicability. Line 472 (track changes).

Refs

- Please see recent Plos One papers, and add pmid numbers.

Response: We added PMID numbers. Line 484 (track changes).

All in all, the authors have submitted a nice draft, considered interesting and easily intelligible. This paper should be worth following after major revisions. 

RESPONSE TO REVIEWERS

REVIEWER 3

Dear reviewer, thanks for your comments that support us to improve the manuscript. We specify the changes and corrections made as follows:

-General - explaining some of the technical terminology would be helpful - I do not understand 'intention to practice' - does this mean intention to prescribe antibiotics within the US guidelines which you have highlighted as the basis for Colombian dentistry?

Response: In the introduction, we added information about the meaning of Awareness, Attitudes, and intention to practice. Line 102. In addition, we specified in the Methods section, subheading “Impact of the VLE on dentist”, “intention to practice” the following: according to the AHA and ADA guidelines, and recent studies Line 194 (manuscript with track changes). 

-Introduction - Too many references - 37 in just 5 paragraphs. Please think really carefully about the best one or two to chose to make each point rather than including up to 5 citations per sentence. For example, your first sentence should include the FDI World Dental Federation's policy statement or white paper on tackling overprescribing of antibiotics in dentistry ahead of many of the other references. And in the opening sentence of your third paragraph, could I suggest a single systematic review which looked at why dentists prescribe antibiotics unnecessarily: https://academic.oup.com/jac/article/74/8/2139/5475276

Response: We reduced the references, including only the relevant studies. Moreover, we incorporated the FDI World Dental Federation's policy statement and the recommended systematic review. Line 73, line 86, line 102, line 115, line 117, line 121 and line 526 (track changes).

-Introduction - para 2 - guidelines for appropriate dental antibiotic prescribing differ around the world so you need to be clear in this paper that you are referring to that which is deemed appropriate in Colombia. And I think you have a typo as periodontal abscesses are less likely to be life-threatening than periapical abscesses. Later in the same paragraph, you state that orthopaedic surgeons 'must' define the antibiotic regimen - this does not make sense given the first part of this sentence.

Response: We modified this paragraph according to the recommendations. Line 79 and line 87 (track changes).

-Introduction - para 3 - other than the first sentence of this paragraph, much of it would sit more comfortably in the discussion section.

Response: We decided not to include this information because, in the methods section, we specified that the course was developed according to the results from our prior study and included the reference. Line 151 (track changes).

-Methods - Be clear about when you undertook this study - this is particularly important as you refer to it in the discussion section.

Response: In methods, we added the time frame of the study. Line 141 (track changes).

-Thinning out the detail to that which is essential will make this section much clearer. Don't repeat information which is already in the tables. 

Response: We deleted information that is contained in the tables. Line 167 (track changes).

-And be careful of jargon: alpha risk and beta risk mean nothing to the general reader. Simply say that your sample size was calculated as 140 dentists based on a cut off for significance of p=0.05.

Response: We changed to “The sample size was calculated as 140 dentists based on a cut-off for significance of p=0.05”. Line 143 (track changes).

-Also, could you mention the date when the ethics committee approved it?

Response: We included the date when the ethics committee approved the study. Line 137 (track changes).

-Table 1 - this needs to be streamlined to get rid of superfluous detail (eg 'welcome message' and 'course detail')

Response: In table 1, we deleted non-relevant information. Line 175 (track changes).

-Results - Again don't repeat information from the tables in the text. For example in the second sentence just say 'Details of the sociodemographic characteristics of participating dentists are included in Table 2.' In spite of its title, you have included more than just sociodemographic characteristics in Table 2. Baseline levels of awareness, attitudes and 'intention to practice' should be in a separate table. And Table 3 has WAY too much detail for the paper. You need to find a way to summarise this into a format digestible by the reader - or put it into an on-line appendix.

Response: We deleted repeated information from the tables and separated the table containing baseline levels of awareness, attitudes, and intention to practice. In addition, we moved Table 3 as a supplemental appendix (S1 Appendix). Line 229, line 235 and line 237 (track changes).

-Discussion - the key messages from the paper need to come through this section much more clearly. Given the extensive use of the US guidelines in the rest of the paper, I was rather surprised that you didn't reference any of the US work on dental antibiotic stewardship from Suda, Durkin or Goff.

Response: We organized the ideas to present them more clearly. Also, we included references about US work on dental antibiotic stewardship from Suda, Durkin, and Goff. Line 599, line 615 and line 439 (track changes).

---

## [Decision Letter · Decision Letter 1]

28 Sep 2021

PONE-D-21-04062R1Impact of a Virtual Learning Environment on the conscious prescription of antibiotics in Colombian dentistsPLOS ONE

Dear Dr. Angarita Díaz,

Thank you for submitting your manuscript to PLOS ONE. After careful consideration, we feel that it has merit but does not fully meet PLOS ONE’s publication criteria as it currently stands. Therefore, we invite you to submit a revised version of the manuscript that addresses the points raised during the review process.

Having intensively re-reviewed your revised draft, our external reviewers again differed with their final recommendations, at least to some extent. Thus, I have double checked your revised version, to come to a more balanced decision (see R #2). All in all, our identified shortcomings are considered reasonable with regard to both PLOS ONE’s quality standards and our readership's expectations. Therefore, we invite you to re-submit a carefully revised version of the manuscript that addresses EACH AND EVERY point raised during the current review process. Please note that more than three reviewed versions is not considered acceptable, and remember that a further non-convincing revision (not considered acceptable with regard to language, content, reviewers' constructive criticism, generalizable conclusions, and/or Authors' Guidelines) must lead to outright reject. 

We look forward to receiving your revised manuscript.

Kind regards,

Prof. Dr. Dr. h. c. Andrej M Kielbassa

Academic Editor

PLOS ONE

Reviewers' comments:

Reviewer's Responses to Questions

**Comments to the Author**

1. If the authors have adequately addressed your comments raised in a previous round of review and you feel that this manuscript is now acceptable for publication, you may indicate that here to bypass the “Comments to the Author” section, enter your conflict of interest statement in the “Confidential to Editor” section, and submit your "Accept" recommendation.

Reviewer #1: All comments have been addressed

Reviewer #2: (No Response)

Reviewer #4: (No Response)

2. Is the manuscript technically sound, and do the data support the conclusions?

Reviewer #1: Yes

Reviewer #2: No

Reviewer #4: Yes

3. Has the statistical analysis been performed appropriately and rigorously? 

Reviewer #1: Yes

Reviewer #2: Yes

Reviewer #4: Yes

4. Have the authors made all data underlying the findings in their manuscript fully available?

Reviewer #1: Yes

Reviewer #2: Yes

Reviewer #4: Yes

5. Is the manuscript presented in an intelligible fashion and written in standard English?

Reviewer #1: Yes

Reviewer #2: Yes

Reviewer #4: Yes

6. Review Comments to the Author

Reviewer #1: (No Response)

Reviewer #2: Abstract

- Please note that "(p=0.000)" is not very probable. Please double check, and revise for "(p < 0.0001)". Please revise thoroughly throughout your text.

- Revise for Journal style. "(p=0.042)" must read "(p = 0.042)".

- Double check for typos, see "(p=0,000)", and revise carefully. Please note that one flawless drafts will be considered ready to proceed, and please note that all 12 co-authors have approved this version, so such typos would not seem understandable.

Intro

- Still, English remains a concern, see "This number is expected to continue increasing (...)." Again, revise carefully.

- Still, a sound null hypothesis is missing at the end of this section. Remember that H0 must be deducible from the foregoing thoughts.

Meths

- Heading must read "Materials and methods". Even if boring, all co-authors are strongly encouraged to stick to https://journals.plos.org/plosone/s/submission-guidelines.

- "The sample size was calculated as 140 dentists based on a cut-off for significance of p=0.05." This would seem hard to follow. Please provide your primary end-point for this cut-off value.

- Again, do not use legal terms with your text, delete "Corp".

Results

- "(...) and baseline levels of awareness, attitudes, and intention to practice are shown in Table 3." Do not leave the reader alone with your Tables. Instead, you must guide the reader with your full text. Without doubling each and every result, please provide the most important outcomes.

- Please double check and revise for correct parentheses, see "(before: 5.0 IQR [4.0-6.0], after 6.0, IQR [5.0-6.0], (p=0.000)", and compare to "(before: 5.0 IQR [4.0-5.25], after 5.0, IQR [4.0-6.0] p=0.000)". Revise carefully for uniform formatting.

- Regarding the p values, see comments given above, and revise carefully.

Disc

- Refer to H0 when starting this section.

Refs

- Again, please revise for uniform formatting. Again, it would seem astonishing why the 12 co-authors simply have ignored the previous recommendations (which would seem easy to follow).

- For example "Sofhauser C. Intention in nursing practice. nursing science quarterly. 2016;29(1):31-34. https://doi.org/10.1177/0894318415614629 PMID: 26660773." must read "Sofhauser C. Intention in nursing practice. Nurs Sci Q. 2016; 29(1): 31–34. https://doi.org/10.1177/0894318415614629 PMID: 26660773" Double check for spacebar use, correct Journal abbreviation, correct hyphen ("–", not "-"), and so on. Remember to re-submit a flawless manuscript, to avoid possible errors with the proofs. This is considered the Co-Authors' task, not the Reviewers' (or the Type-setter's) one.

In total, this revised and re-submitted draft has been considerably improved, but is not ready to proceed.

Reviewer #4: This paper is about the prescription of antimicrobials in dentistry, and describes the implementation of a VLE course and it´s impacts on awareness, attitudes and intention to practice regarding antibiotic prescription in dentistry. In the face of the challenge of increasing microbial resistance, this is an important topic, and successful experiences must be reported. Thus, I think this article is worthy of publication. However, I have a few comments that might help to improve the quality of the article before publication. I will refer to the page and line numbers in the PDF version of the Revised Manuscript with Track Changes.

ABSTRACT:

- Line 45: the authors correctly followed the recommendation of reviewer 2 and modified the P values on a 3-digit basis. However, the notation when the value of p is less than 0.001 must be p<0.001, and not p=0.000. I suggest this modification throughout the article.

INTRODUCTION:

- Line 129: although authors revised the text in response to reviewer 2's indication that a sound null hypothesis is missing, this H0 was not really clear. Perhaps a proposal is: "considering that the VLE is a teaching-learning method that may not improve awareness, attitudes, etc., we develop a VLE and measure the impact". I believe that the way of writing can make the null hypothesis clearer.

METHODS:

- Line 199: the score categorization used follows a certain logic for awareness and attitudes (high level being considered only if there is total correctness) and another for intention to practice (high level for those who get around 63 to 100% of the questions right). For awareness and attitudes, getting 63% of the questions right would mean low or medium level. I would like to understand the authors' option for this categorization, and, if they deem it pertinent, I suggest putting this as a limitation of the study in the discussion.

RESULTS:

- Lines 250 and 257: figures 2 and 3 always compare 4 sets of data, referring to "before", "immediate", another "before" and "6 months". In my opinion, the occurrence of 2 "before" is only clear when analyzing the supporting information. Thus, it may be worth clarifying in Methods that the comparisons were made with the total sample that completed the VLE course (first "before" and "immediate" sample) and with those who answered the questionnaire after 6 months (second "before" and "6 months" sample).

DISCUSSION:

- Line 438: revise English in the sentence "A study in the medical field, in which an online course was given to students and showed that >90% of the participants had a good score on activities related to basic knowledge (principles of antibiotic stewardship);".

- Line 460: the limitations paragraph is quite modest, associating the limitations mainly with COVID-19. In addition, the authors sticked a final paragraph of discussion on limitations. I suggest a distinct paragraph, inserting the following topics: 1. study design, without a control group; 2. score categorization (already indicated above in Methods, Line 199); and 3. indicate as selection bias the fact that dentists from a previous study were invited to participate in this study.

7. PLOS authors have the option to publish the peer review history of their article (what does this mean?). If published, this will include your full peer review and any attached files.

Reviewer #1: No

Reviewer #2: No

Reviewer #4: **Yes: **Michel Laks

---

## [Author Response · Author response to Decision Letter 1]

8 Nov 2021

Dear Editor:

Prof. Dr. Andrej M Kielbassa

Academic Editor

PLOS ONE

Please, in the attached file, you will find the revised manuscript PONE-D-21-04062: " Impact of a Virtual Learning Environment on the conscious prescription of antibiotics in Colombian dentists" by Angarita Diaz et al., that we are submitting for publication in "PLOS ONE". We have added responses to the suggestions and comments of the reviewers and would like to thank them for all their time and consideration of our work.

Best regards,

The authors

RESPONSE JOURNAL REQUIREMENTS:

Reviewer #2: 

Dear reviewer, thanks for your comments that support us to improve the manuscript. We specify the changes and corrections made as follows:

Abstract

- Please note that "(p=0.000)" is not very probable. Please double check, and revise for "(p < 0.0001)". Please revise thoroughly throughout your text.

Response: We modified according to your suggestion, and to the guide: “P-values. Report exact p-values for all values greater than or equal to 0.001. P-values less than 0.001 may be expressed as p < 0.001”.

- Revise for Journal style. "(p=0.042)" must read "(p = 0.042)".

Response: We put the corresponding spaces.

- Double check for typos, see "(p=0,000)", and revise carefully. Please note that one flawless drafts will be considered ready to proceed, and please note that all 12 co-authors have approved this version, so such typos would not seem understandable.

Response: We corrected the typo mistakes.

Intro

- Still, English remains a concern, see "This number is expected to continue increasing (...)." Again, revise carefully.

Response: We modified “This number is expected to increase up to 10 million deaths per year by 2050, as anticipated by the British government”. Line 57 (manuscript with track changes). We recheck and improve the English of the entire document.

- Still, a sound null hypothesis is missing at the end of this section. Remember that H0 must be deducible from the foregoing thoughts.

Response: We added the null hypothesis: The null hypothesis of the study is that the development and implementation of a VLE on conscious antibiotic prescription in dentistry does not have a favorable impact on participants’ awareness, attitudes, and intention to practice. Line 121 (manuscript with track changes)

Meths

- Heading must read "Materials and methods". Even if boring, all co-authors are strongly encouraged to stick to https://journals.plos.org/plosone/s/submission-guidelines.

Response: We modified the mistake with the heading. Line 126 (manuscript with track changes)

- "The sample size was calculated as 140 dentists based on a cut-off for significance of p=0.05." This would seem hard to follow. Please provide your primary end-point for this cut-off value.

Response: We added information about the sample size: “The sample size was calculated for each city including dentists with a Dentistry degree until 2016 according to the last update from the Observatorio Laboral para la Educación through a paired sample t-test (repeated measures). By accepting an alpha risk of 0.05 (5%) and a beta risk of 0.1 (10% potential) in a bilateral contrast, a total of 140 dentists were required. A 15% loss to follow-up was estimated.” Line 136 (manuscript with track changes)

However, in this document, we specify that reviewer # 3 on may, wrote: Just say that your sample size was calculated as 140 dentists based on a significance cutoff of p = 0.05. For that reason, we had deleted the information on the before copy.

- Again, do not use legal terms with your text, delete "Corp".

Response: We deleted “Corp”. Line 213 126 (manuscript with track changes).

Results

- "(...) and baseline levels of awareness, attitudes, and intention to practice are shown in Table 3." Do not leave the reader alone with your Tables. Instead, you must guide the reader with your full text. Without doubling each and every result, please provide the most important outcomes.

Response: We included: “Most of the participants were female (69.9%), performed a private clinical practice (60.2%), and had more than 10 years of clinical experience (45.6%); dentists with or without a postgraduate formation were equal (50%) (Table 2). Furthermore, most dentists showed a medium level of awareness (57.3%) and attitudes (80.1%) and a high level of intention to practice (91.3%) (Table 3)” Line 230 (manuscript with track changes).

In this document, we specify that reviewer # 3 in the correction above, wrote: Again don't repeat information from the tables in the text. For example in the second sentence just say 'Details of the sociodemographic characteristics of participating dentists are included in Table 2.' For that reason, we had deleted the information on the before copy.

- Please double check and revise for correct parentheses, see "(before: 5.0 IQR [4.0-6.0], after 6.0, IQR [5.0-6.0], (p=0.000)", and compare to "(before: 5.0 IQR [4.0-5.25], after 5.0, IQR [4.0-6.0] p=0.000)". Revise carefully for uniform formatting.

Response: We compared and standardized the information.

- Regarding the p values, see comments given above, and revise carefully.

Response: We modified according to the guide and the reviewer.

Disc

- Refer to H0 when starting this section.

Response: We started the Discussion section with H0: “The findings from this study allowed us to reject the null hypothesis as significant differences in awareness and intention to practice of participants on antibiotic prescription were found”. Line 372 (manuscript with track changes).

Refs

- Again, please revise for uniform formatting. Again, it would seem astonishing why the 12 co-authors simply have ignored the previous recommendations (which would seem easy to follow).

Response: We modified according to the guide (which request doi) and recent Dentistry-Plos One papers (which request pmid).

- For example "Sofhauser C. Intention in nursing practice. nursing science quarterly. 2016;29(1):31-34. https://doi.org/10.1177/0894318415614629 PMID: 26660773." must read "Sofhauser C. Intention in nursing practice. Nurs Sci Q. 2016; 29(1): 31–34. https://doi.org/10.1177/0894318415614629 PMID: 26660773" Double check for spacebar use, correct Journal abbreviation, correct hyphen ("–", not "-"), and so on. Remember to re-submit a flawless manuscript, to avoid possible errors with the proofs. This is considered the Co-Authors' task, not the Reviewers' (or the Type-setter's) one.

In total, this revised and re-submitted draft has been considerably improved, but is not ready to proceed.

Response: We did check for spacebar use, corrected the journal abbreviation, and fixed the hyphen. Also, modified according to the guide, which request doi: “A DOI number for the full-text article is acceptable as an alternative to or in addition to traditional volume and page numbers. When providing a DOI, adhere to the format in the example above with both the label and full DOI included at the end of the reference (doi: 10.1016/j.molimm.2014.11.005). Do not provide a shortened DOI or the URL.”

Reviewer #4: 

Dear reviewer, thanks for your comments that support us to improve the manuscript. We specify the changes and corrections made as follows:

This paper is about the prescription of antimicrobials in dentistry, and describes the implementation of a VLE course and it´s impacts on awareness, attitudes and intention to practice regarding antibiotic prescription in dentistry. In the face of the challenge of increasing microbial resistance, this is an important topic, and successful experiences must be reported. Thus, I think this article is worthy of publication. However, I have a few comments that might help to improve the quality of the article before publication. I will refer to the page and line numbers in the PDF version of the Revised Manuscript with Track Changes.

ABSTRACT:

- Line 45: the authors correctly followed the recommendation of reviewer 2 and modified the P values on a 3-digit basis. However, the notation when the value of p is less than 0.001 must be p<0.001, and not p=0.000. I suggest this modification throughout the article.

Response: We modified according to your suggestion and, to the guide: “P-values. Report exact p-values for all values greater than or equal to 0.001. P-values less than 0.001 may be expressed as p < 0.001”. Line 42 (manuscript with track changes).

INTRODUCTION:

- Line 129: although authors revised the text in response to reviewer 2's indication that a sound null hypothesis is missing, this H0 was not really clear. Perhaps a proposal is: "considering that the VLE is a teaching-learning method that may not improve awareness, attitudes, etc., we develop a VLE and measure the impact". I believe that the way of writing can make the null hypothesis clearer.

Response: We added the null hypothesis: “The null hypothesis of the study is that the development and implementation of a VLE on conscious antibiotic prescription in dentistry does not have a favorable impact on participants’ awareness, attitudes, and intention to practice.” Line 121 (manuscript with track changes).

METHODS:

- Line 199: the score categorization used follows a certain logic for awareness and attitudes (high level being considered only if there is total correctness) and another for intention to practice (high level for those who get around 63 to 100% of the questions right). For awareness and attitudes, getting 63% of the questions right would mean low or medium level. I would like to understand the authors' option for this categorization, and, if they deem it pertinent, I suggest putting this as a limitation of the study in the discussion.

Response: We added and explained the limitation: “….score categorization was a limitation as according to the percentile classification during the questionnaire validation, a high level of awareness and attitudes was achieved with 100% of correct answers whereas, for intention to practice, a high level was reached with 63% - 100% of correct answers [27]. Line 466 (manuscript with track changes).

RESULTS:

- Lines 250 and 257: figures 2 and 3 always compare 4 sets of data, referring to "before", "immediate", another "before" and "6 months". In my opinion, the occurrence of 2 "before" is only clear when analyzing the supporting information. Thus, it may be worth clarifying in Methods that the comparisons were made with the total sample that completed the VLE course (first "before" and "immediate" sample) and with those who answered the questionnaire after 6 months (second "before" and "6 months" sample).

Response: In Materials and Methods, we clarified about your suggestion: The immediate impact (“immediately after” intervention) was compared to the first measurement (“before” intervention) of the total sample of dentists that completed the course. To analyze information retention, results from dentists that answered the questionnaire at 6 months (“6 months” post-intervention) were compared to the data obtained from the same population that answered the initial questionnaire (“before” intervention). Line 175 (manuscript with track changes).

DISCUSSION:

- Line 438: revise English in the sentence "A study in the medical field, in which an online course was given to students and showed that >90% of the participants had a good score on activities related to basic knowledge (principles of antibiotic stewardship);".

Response: We modified the sentence “In a similar study, an online course given to 310 medical students showed that the majority had an average score of 90% on activities related to basic knowledge (principles of antibiotic stewardship). Moreover, 153 of these participants completed all five interactive online clinical cases, with about half of them scoring over 90% in the first attempt [36]”. Line 441 (manuscript with track changes).

- Line 460: the limitations paragraph is quite modest, associating the limitations mainly with COVID-19. In addition, the authors sticked a final paragraph of discussion on limitations. I suggest a distinct paragraph, inserting the following topics: 1. study design, without a control group; 2. score categorization (already indicated above in Methods, Line 199); and 3. indicate as selection bias the fact that dentists from a previous study were invited to participate in this study.

Response: We organized the limitation according to your suggestion. Line 462 (manuscript with track changes).

---

## [Decision Letter · Decision Letter 2]

23 Nov 2021

PONE-D-21-04062R2Impact of a Virtual Learning Environment on the conscious prescription of antibiotics in Colombian dentistsPLOS ONE

Dear Dr. Angarita Díaz,

Thank you for submitting your manuscript to PLOS ONE. After careful consideration, we feel that it has merit but does not fully meet PLOS ONE’s publication criteria as it currently stands. Therefore, we invite you to submit a revised version of the manuscript that addresses the points raised during the review process.

Having intensively reviewed your revised draft, our external reviewers basically have agreed with their final recommendations. Additionally, I have double checked your submitted version, to come to a more balanced decision (see R #2). All in all, I am convinced that your revised paper will be worth following, even if your revised version still would benefit from thorough re-edits and language polishing.

We look forward to receiving your revised manuscript.

Kind regards,

Andrej M Kielbassa

Academic Editor

PLOS ONE

Journal Requirements:

Reviewers' comments:

Reviewer's Responses to Questions

**Comments to the Author**

1. If the authors have adequately addressed your comments raised in a previous round of review and you feel that this manuscript is now acceptable for publication, you may indicate that here to bypass the “Comments to the Author” section, enter your conflict of interest statement in the “Confidential to Editor” section, and submit your "Accept" recommendation.

Reviewer #1: All comments have been addressed

Reviewer #2: All comments have been addressed

Reviewer #4: All comments have been addressed

2. Is the manuscript technically sound, and do the data support the conclusions?

Reviewer #1: Yes

Reviewer #2: Yes

Reviewer #4: Yes

3. Has the statistical analysis been performed appropriately and rigorously? 

Reviewer #1: Yes

Reviewer #2: Yes

Reviewer #4: Yes

4. Have the authors made all data underlying the findings in their manuscript fully available?

Reviewer #1: Yes

Reviewer #2: Yes

Reviewer #4: Yes

5. Is the manuscript presented in an intelligible fashion and written in standard English?

Reviewer #1: Yes

Reviewer #2: Yes

Reviewer #4: Yes

6. Review Comments to the Author

Reviewer #1: Dear authors,

Some concerns remained and requested minor revision.

Material and Methods

I suggest adjusting the text:

l. 128 “The present study is minimal risk research and was approved….”

for this phrase, “This study was approved….”

The calculus of sample size remains unclear. A total of 140 dentists were required, but how many dentists per city were needed, considering the total number of dentists per city?

Results

l. 230 “A total of 206 dentists completed the course.”

How many participants initiate the course? Please, write about the loss of the participants. It is essential to know how many participants initiated the course and how many completed the course.

I suggest complementing the legend’s information of the Tables:

l. 237 Table 2. Sociodemographic characteristics of the participants (n = 206)

l. 239 Table 3. Baseline levels of awareness, attitudes, and intention to practice of the participants (n = 206)

The first column of Table 2 and Table 3 indicates the Absolute Number and not Frequency. The Frequency means prevalence(percentage).

l. 232 “Furthermore, most dentists showed a medium level of awareness (57.3%) and attitudes (80.1%) and a high level of intention to practice (91.3%) (Table 3).” Does 80.1% represent a medium level?

Why was the Wilcoxon signed-rank test used instead of the Friedman test? The comparison was made among three groups (basal, immediately after, and 6 -months post-intervention) and not only two groups.

According to the instructions, the authors must reorder:

• Acknowledgments

• References

• Supporting information captions

Reviewer #2: The Co-Authors have satisfyingly modified their draft, and most comments have been followed. This revised and re-submitted manuscript is considered ready to proceed.

Reviewer #4: (No Response)

7. PLOS authors have the option to publish the peer review history of their article (what does this mean?). If published, this will include your full peer review and any attached files.

Reviewer #1: No

Reviewer #2: No

Reviewer #4: **Yes: **Michel Laks

---

## [Author Response · Author response to Decision Letter 2]

23 Dec 2021

Dear Editor:

Prof. Dr. Andrej M Kielbassa

Academic Editor

PLOS ONE

Please, in the attached file, you will find the revised manuscript PONE-D-21-04062: " Impact of a Virtual Learning Environment on the conscious prescription of antibiotics in Colombian dentists" by Angarita Diaz et al., that we are submitting for publication in "PLOS ONE". We have added responses to the suggestions and comments of the reviewers and would like to thank them for all their time and consideration of our work.

Moreover, according to the comment, “I am convinced that your revised paper will be worth following, even if your revised version still would benefit from thorough re-edits and language polishing.”, we performed the editing process on this occasion and added the certification. During the edition, one word of the manuscript's title changed: Impact of a Virtual Learning Environment on the conscious prescription of antibiotics among Colombian dentists"

In relation to the comment, “Please review your reference list to ensure that it is complete and correct. If you have cited papers that have been retracted, please include the rationale for doing so in the manuscript text, or remove these references and replace them with relevant current references. Any changes to the reference list should be mentioned in the rebuttal letter that accompanies your revised manuscript. If you need to cite a retracted article, indicate the article’s retracted status in the References list and also include a citation and full reference for the retraction notice”. We thoroughly reviewed the reference list.

Best regards,

The authors

RESPONSE JOURNAL REQUIREMENTS:

Reviewer #1: 

Dear reviewer, thanks for your comments that support us to improve the manuscript. We specify the changes and corrections made as follows:

Some concerns remained and requested minor revision.

Material and Methods

I suggest adjusting the text:

l. 128 “The present study is minimal risk research and was approved….”

for this phrase, “This study was approved….”

Response: We modified this phrase according to your suggestion. Line 126. (manuscript with track changes)

The calculus of sample size remains unclear. A total of 140 dentists were required, but how many dentists per city were needed, considering the total number of dentists per city?

Response: We specified the total number of required dentists per city: “At least 20 dentists per city registered at the local Health Secretary or the Sistema Integrado de Información de la Protección Social of the Ministry of Health were required.” Line 138. (manuscript with track changes)

Results

l. 230 “A total of 206 dentists completed the course.”

How many participants initiate the course? Please, write about the loss of the participants. It is essential to know how many participants initiated the course and how many completed the course.

Response: We added the number of participants that initiated the course and how many completed the course: “A total of 279 dentists initiated the VLE, of which 73.8% completed the course.” Line 227. (manuscript with track changes). We would like to specify that we did not ask about the reasons for course dropout in the study.

I suggest complementing the legend’s information of the Tables:

l. 237 Table 2. Sociodemographic characteristics of the participants (n = 206)

Response: We modified this legend according to your suggestion. Line 235. (manuscript with track changes)

l. 239 Table 3. Baseline levels of awareness, attitudes, and intention to practice of the participants (n = 206)

Response: We modified this legend according to your suggestion. Line 237. (manuscript with track changes)

The first column of Table 2 and Table 3 indicates the Absolute Number and not Frequency. The Frequency means prevalence(percentage).

Response: We modified this column according to your suggestion.

l. 232 “Furthermore, most dentists showed a medium level of awareness (57.3%) and attitudes (80.1%) and a high level of intention to practice (91.3%) (Table 3).” Does 80.1% represent a medium level?

Response: We modified the redaction as follows: “Furthermore, 57.3% and 80.1% of dentists showed a medium level of awareness and attitudes, respectively, and 91.3% had a high level of intention to practice practice (Table 3).” Line 231. (manuscript with track changes).

Why was the Wilcoxon signed-rank test used instead of the Friedman test? The comparison was made among three groups (basal, immediately after, and 6 -months post-intervention) and not only two groups.

Response: The comparisons were between two groups. Line 169 specified this information regarding the comparisons: “To determine the effect of the VLE on dentists, participants answered a questionnaire intended to measure their levels of awareness, attitudes, and intention to practice antibiotic prescription before, immediately after, and 6 months post-intervention. The immediate impact (“immediately after” intervention) was compared to the first measurement (“before” intervention) of the total sample of dentists who completed the course. To analyze information retention, results from dentists who answered the questionnaire at 6 months (“6 months” post-intervention) were compared to the data obtained from the same population that answered the initial questionnaire (“before” intervention).”

According to the instructions, the authors must reorder:

• Acknowledgments

• References

• Supporting information captions

Response: We re-ordered the sections.

---

## [Decision Letter · Decision Letter 3]

5 Jan 2022

Impact of a Virtual Learning Environment on the conscious prescription of antibiotics among Colombian dentists

PONE-D-21-04062R3

Dear Dr. Angarita Díaz,

We’re pleased to inform you that your manuscript has been judged scientifically suitable for publication and will be formally accepted for publication once it meets all outstanding technical requirements. Congratulations!

Kind regards, and stay healthy, please

Andrej M Kielbassa, Prof. Dr. med. dent. Dr. h. c.

Academic Editor

PLOS ONE

Additional Editor Comments (optional):

Reviewers' comments:

Reviewer's Responses to Questions

**Comments to the Author**

1. If the authors have adequately addressed your comments raised in a previous round of review and you feel that this manuscript is now acceptable for publication, you may indicate that here to bypass the “Comments to the Author” section, enter your conflict of interest statement in the “Confidential to Editor” section, and submit your "Accept" recommendation.

Reviewer #2: All comments have been addressed

2. Is the manuscript technically sound, and do the data support the conclusions?

Reviewer #2: Yes

3. Has the statistical analysis been performed appropriately and rigorously? 

Reviewer #2: Yes

4. Have the authors made all data underlying the findings in their manuscript fully available?

Reviewer #2: Yes

5. Is the manuscript presented in an intelligible fashion and written in standard English?

Reviewer #2: Yes

6. Review Comments to the Author

Reviewer #2: This revised and re-submitted manuscript has been consididerably improved, and would seem ready to proceed.

7. PLOS authors have the option to publish the peer review history of their article (what does this mean?). If published, this will include your full peer review and any attached files.

Reviewer #2: No

---

## [Editor Report · Acceptance letter]

19 Jan 2022

PONE-D-21-04062R3 

Impact of a Virtual Learning Environment on the conscious prescription of antibiotics among Colombian dentists 

Dear Dr. Angarita-Díaz:

I'm pleased to inform you that your manuscript has been deemed suitable for publication in PLOS ONE. Congratulations! Your manuscript is now with our production department. 

Kind regards, 

on behalf of

Prof. Dr. med. dent. Dr. h. c. Andrej M Kielbassa 

Academic Editor

PLOS ONE